# Learning to Propagate for Graph Meta-Learning

**Lu Liu[1], Tianyi Zhou[2], Guodong Long[1], Jing Jiang[1], Chengqi Zhang[1]**
[1]Center for Artificial Intelligence, University of Technology Sydney
[2]Paul G. Allen School of Computer Science & Engineering, University of Washington
lu.liu-10@student.uts.edu.au, tianyizh@uw.edu, guodong.long@uts.edu.au
jing.jiang@uts.edu.au, chengqi.zhang@uts.edu.au

## Abstract

Meta-learning extracts the common knowledge from learning different tasks and uses it for unseen tasks. It can significantly improve tasks that suffer from insufficient training data, e.g., few-shot learning. In most meta-learning methods, tasks are implicitly related by sharing parameters or optimizer. In this paper, we show that a meta-learner that explicitly relates tasks on a graph describing the relations of their output dimensions (e.g., classes) can significantly improve few-shot learning. The graph's structure is usually free or cheap to obtain but has rarely been explored in previous works. We develop a novel meta-learner of this type for prototype based classification, in which a prototype is generated for each class, such that the nearest neighbor search among the prototypes produces an accurate classification. The meta-learner, called "Gated Propagation Network (GPN)", learns to propagate messages between prototypes of different classes on the graph, so that learning the prototype of each class benefits from the data of other related classes. In GPN, an attention mechanism aggregates messages from neighboring classes of each class, with a gate choosing between the aggregated message and the message from the class itself. We train GPN on a sequence of tasks from many-shot to few-shot generated by subgraph sampling. During training, it is able to reuse and update previously achieved prototypes from the memory in a life-long learning cycle. In experiments, under different training-test discrepancy and test task generation settings, GPN outperforms recent meta-learning methods on two benchmark datasets. The code of GPN and dataset generation is available at `https://github.com/liulu112601/Gated-Propagation-Net`.

## 1 Introduction

The success of machine learning (ML) during the past decade has relied heavily on the rapid growth of computational power, new techniques training larger and more representative neural networks, and critically, the availability of enormous amounts of annotated data. However, new challenges have arisen with the move from cloud computing to edge computing and Internet of Things (IoT), and demands for customized models and local data privacy are increasing, which raise the question: how can a powerful model be trained for a specific user using only a limited number of local data? Meta-learning, or "learning to learn", can address this few-shot challenge by training a shared meta-learner model on top of distinct learner models for implicitly related tasks. The meta-learner aims to extract the common knowledge of learning different tasks and adapt it to unseen tasks in order to accelerate their learning process and mitigate their lack of training data. Intuitively, it allows new learning tasks to leverage the "experiences" from the learned tasks via the meta-learner, though these tasks do not directly share data or targets.

Meta-learning methods have demonstrated clear advantages on few-shot learning problems in recent years. The form of a meta-learner model can be a similarity metric (for K-nearest neighbor (KNN)

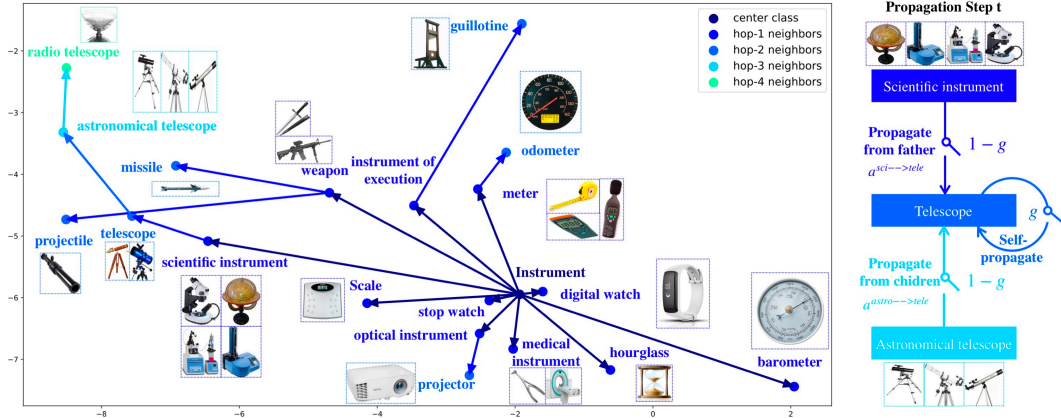

Figure 1: **LEFT:** t-SNE [17] visualization of the class prototypes produced by GPN and the associated graph. **RIGHT:** GPN's propagation mechanism for one step: for each node, its neighbors pass messages (their prototypes) to it according to attention weight $a$, where a gate further choose to accept the message from the neighbors $g^+$ or from the class itself $g^*$.

classification in each task) [24], a shared embedding module [19], an optimization algorithm [14] or parameter initialization [5], and so on. If a meta-learner is trained on sufficient and different tasks, it is expected to be generalized to new and unseen tasks drawn from the same distribution as the training tasks. Thereby, different tasks are related via the shared meta-learner model, which implicitly captures the shared knowledge across tasks. However, in a lot of practical applications, the relationships between tasks are known in the form of a graph of their output dimensions, for instance, species in the biology taxonomy, diseases in the classification coding system, and merchandise on an e-commerce website.

In this paper, we study the meta-learning for few-shot classification tasks defined on a given graph of classes with mixed granularity, that is, the classes in each task could be an arbitrary combination of classes with different granularity or levels in a hierarchical taxonomy. The tasks can be classification of cat vs mastiff (dog) or an m-vs-rest task, e.g. classification that aims to distinguish among cat, dog and others. In particular, we define the graph with each class as a node and each edge connecting a class to its sub-class (i.e., children class) or parent class. In practice, the graph is usually known in advance or can be easily extracted from a knowledge base, such as the WordNet hierarchy for classes in ImageNet [4]. Given the graph, each task is associated with a subset of nodes on the graph. Hence, tasks can be related through the paths on the graph that links their nodes even when they share few output classes. In this way, different tasks can share knowledge by message passing on the graph.

We develop Gated Propagation Network (GPN) to learn how to pass messages between nodes (i.e., classes) on the graph for more effective few-shot learning and knowledge sharing. We use the setting from [24]: given a task, the meta-learner generates a prototype representing each class by using only few-shot training data, and during test a new sample is classified to the class of its nearest prototype. Hence, each node/class is associated with a prototype. Given the graph structure, we let each class send its prototype as a message to its neighbors, while a class received multiple messages needs to combine them with different weights and update its prototype accordingly. GPN learns an attention mechanism to compute the combination weights and a gate to filter the message from different senders (which also includes itself). Both the attention and gate modules are shared across tasks and trained on various few-shot tasks, so they can be generalized to the whole graph and unseen tasks. Inspired by the hippocampal memory replay mechanism in [2] and its application in reinforcement learning [18], we also retain a memory pool of prototypes per training class, which can be reused as a backup prototype for classes without training data in future tasks.

We evaluate GPN under various settings with different distances between training and test classes, different task generation methods, and with or without class hierarchy information. To study the effects of distance (defined as the number of hops between two classes) between training and test classes, we extract two datasets from *tiered*ImageNet [22]: *tiered*ImageNet-Far and *tiered*ImageNet-Close. To evaluate the model's generalization performance, test tasks are generated by two subgraph sampling methods, i.e., random sampling and snowball sampling [8] (snowball sampling can restrict the distance of the targeted few-shot classes). To study whether/when the graph structure is more helpful,

we evaluate GPN with and without using class hierarchy. We show that GPN outperforms four recent few-shot learning methods. We also conduct a thorough analysis of different propagation settings. In addition, the "learning to propagate" mechanism can be potentially generalized to other fields.

## 2 Related Works

Meta-learning has been proved to be effective on few-shot learning tasks. It trains a meta learner using augmented memory [23, 11], metric learning [27, 24, 3] or learnable optimization [5]. For example, prototypical network [24] applied a distance-based classifier in a trained feature space. We can extend the single prototype per class to an adaptive number of prototypes by infinite mixture model [1]. The feature space could be further improved by scaling features according to different tasks [19]. Our method is built on prototypical network and improves the prototype per class by propagation between prototypes of different classes. Our work also relates to memory-based approaches, in which feature-label pairs are selected into memory by dedicated reading and writing mechanisms. In our case, the memory stores prototypes and improves the propagation efficiency. Auxiliary information, such as unlabeled data [22] and weakly-labeled data [15] has been used to embrace the few-shot challenge. In this paper, we improve the quality of prototype per class by sending messages between prototypes on a graph describing the class relationships.

Our idea of prototype propagation is inspired by belief propagation [20, 28], message passing and label propagation [30, 29]. It is also related to Graph Neural Networks (GNN) [10, 26], which applies convolution/attention iteratively on a graph to achieve node embedding. In contrast, the graph in our paper is a computational graph in which every node is associated with a prototype produced by an CNN rather than a non-parameterized initialization in GNN. Our goal is to obtain a better prototype representation for classes in few-shot classification. Propagation has been applied in few-shot learning for label propagation [16] in a transductive setting to infer the entire query set from support set at once.

## 3 Graph Meta-Learning

### 3.1 Problem Setup

We study "graph meta-learning" for few-shot learning tasks, where each task is associated with a prediction space defined by a subset of nodes on a given graph, e.g., 1) for classification tasks: a subset of classes from a hierarchy of classes; 2) for regression tasks: a subset of variables from a graphical model as the prediction targets; or 3) for reinforcement learning tasks: a subset of actions (or a subsequence of actions). In real-world problems, the graph is usually free or cheap to achieve and can provide weakly-supervised information for a meta-learner since it relates different

Table 1: Notations used in this paper.

| Notation | Definition |
| --- | --- |
| $\mathcal{Y}$ | Ground set of classes for all possible tasks |
| $\mathcal{G} = (\mathcal{Y}, E)$ | Category graph with nodes $\mathcal{Y}$ and edges $E$ |
| $\mathcal{N}_y$ | The set of neighbor classes of $y$ on graph $\mathcal{G}$ |
| $\mathcal{M}(\cdot; \Theta)$ | A meta-learner model with paramter $\Theta$ |
| $T$ | A few-shot classification task |
| $\mathcal{T}$ | Distribution that each task $T$ is drawn from |
| $\mathcal{Y}^T \subseteq \mathcal{Y}$ | The set of output classes in task $T$ |
| $(\boldsymbol{x}, y)$ | A sample with input data $\boldsymbol{x}$ and label $y$ |
| $D^T$ | Distribution of $(\boldsymbol{x}, y)$ in task $T$ |
| $\boldsymbol{P}_y$ | Final output prototype of class $y$ |
| $\boldsymbol{P}_y^t$ | Prototype of class $y$ at step $t$ |
| $\boldsymbol{P}_{y \to y}^t$ | Message sent from class $y$ to itself |
| $\boldsymbol{P}_{\mathcal{N}_y \to y}^t$ | Message sent to class $y$ from its neighbors |

tasks' output spaces via the edges and paths on the graph. However, it has been rarely considered in previous works, most of which relate tasks via shared representation or metric space.

In this paper, we will study graph meta-learning for few-shot classification. In this problem, we are given a graph with nodes as classes and edges connecting each class to its parents and/or children classes, and each task aims to learn a classifier categorizing an input sample into a subset of $N$ classes given only $K$ training samples per class. Comparing to the traditional setting for few-shot classification, the main challenge of graph meta-learning comes from the mixed granularity of the classes, i.e., a task might aim to classify a mixed subset containing both fine and coarse categories. Formally, given a directed acyclic graph (DAG) $\mathcal{G} = (\mathcal{Y}, E)$, where $\mathcal{Y}$ is the ground set of classes for all possible tasks, each node $y \in \mathcal{Y}$ denotes a class, and each directed edge (or arc) $y_i \to y_j \in E$ connects a parent class $y_i \in \mathcal{Y}$ to one of its child classes $y_j \in \mathcal{Y}$ on the graph $\mathcal{G}$. We assume that each learning task $T$ is defined by a subset of classes $\mathcal{Y}^T \subseteq \mathcal{Y}$ drawn from a certain distribution $\mathcal{T}(\mathcal{G})$ defined on the graph, our goal is to learn a meta-learner $\mathcal{M}(\cdot; \Theta)$ that is parameterized by $\Theta$ and can produce a learner model $\mathcal{M}(T; \Theta)$ for each task $T$. This problem can then be formulated by

the following risk minimization of "learning to learn":

$$\min_{\Theta} \mathbb{E}_{T \sim \mathcal{T}(\mathcal{G})} \left[ \mathbb{E}_{(\boldsymbol{x},y) \sim \mathcal{D}^T} - \log \Pr(y|\boldsymbol{x}; \mathcal{M}(T, \Theta))) \right], \tag{1}$$

where $\mathcal{D}^T$ is the distribution of data-label pair $(\boldsymbol{x}, y)$ for a task $T$. In few-shot learning, we assume that each task $T$ is an $N$-way-$K$-shot classification over $N$ classes $\mathcal{Y}^T \subseteq \mathcal{Y}$, and we only observe $K$ training samples per class. Due to the data deficiency, conventional supervised learning usually fails.

We further introduce the form of $\Pr(y|\boldsymbol{x}; \mathcal{M}(T; \Theta))$ in Eq. (1). Inspired by [24], each classifier $\mathcal{M}(T; \Theta)$, as a learner model, is associated with a subset of prototypes $\boldsymbol{P}_{\mathcal{Y}^T}$ where each prototype $\boldsymbol{P}_y$ is a representation vector for class $y \in \mathcal{Y}^T$. Given a sample $\boldsymbol{x}$, $\mathcal{M}(T; \Theta)$ produces the probability of $\boldsymbol{x}$ belonging to each class $y \in \mathcal{Y}^T$ by applying a soft version of KNN: the probability is computed by an RBF Kernel over the Euclidean distances between $f(\boldsymbol{x})$ and prototype $\boldsymbol{P}_y$, i.e.,

$$\Pr(y|\boldsymbol{x}; \mathcal{M}(T; \Theta)) \triangleq \frac{\exp(-\|f(\boldsymbol{x}) - \boldsymbol{P}_y\|^2)}{\sum_{z \in \mathcal{Y}^T} \exp(-\|f(\boldsymbol{x}) - \boldsymbol{P}_z\|^2)}, \tag{2}$$

where $f(\cdot)$ is a learnable representation model for input $\boldsymbol{x}$. The main idea of graph meta-learning is to improve the prototype of each class in $\boldsymbol{P}$ by assimilating their neighbors' prototypes on the graph $\mathcal{G}$. This can be achieved by allowing classes on the graph to send/receive messages to/from neighbors and modify their prototypes. Intuitively, two classes should have similar prototypes if they are close to each other on the graph. Meanwhile, they should not have exactly the same prototype since it leads to large errors on tasks containing both the two classes. The remaining questions are 1) how to measure the similarity of classes on graph $\mathcal{G}$? 2) how to relate classes that are not directly connected on $\mathcal{G}$? 3) how to send messages between classes and how to aggregate the received messages to update prototypes? 4) how to distinguish classes with similar prototypes?

## 3.2 Gated Propagation Network

We propose *Gated Propagation Network* (GPN) to address the graph meta-learning problem. GPN is a meta-learning model that learns how to send and aggregate messages between classes in order to generate class prototypes that result in high KNN prediction accuracy across different $N$-way-$K$-shot classification tasks. Technically, we deploy a multi-head dot-product attention mechanism to measure the similarity between each class and its neighbors on the graph, and use the obtained similarities as weights to aggregate the messages (prototypes) from its neighbors. In each head, we apply a gate to determine whether to accept the aggregated messages from the neighbors or the message from itself. We apply the above propagation on all the classes (together with their neighbors) for multiple steps, so we can relate the classes not directly connected in the graph. We can also avoid identical prototypes of different classes due to the capability of rejecting messages from any other classes except the one from the class itself. In particular, given a task $T$ associated with a subset of classes $\mathcal{Y}^T$ and an $N$-way-$K$-shot training set $\mathcal{D}^T$. At the very beginning, we compute an initial prototype for each class $y \in \mathcal{Y}^T$ by averaging over all the $K$-shot samples belonging to class $y$ as in [24], i.e.,

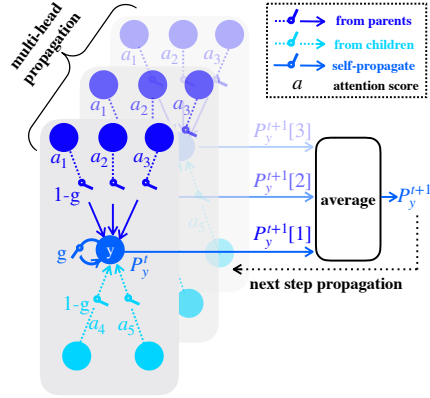

Figure 2: Prototype propagation in GPN: in each step $t + 1$, each class $y$ aggregates prototypes from its neighbors (parents and children) by multi-head attention, and chooses between the aggregated message or the message from itself by a gate $g$ to update its prototype.

$$\boldsymbol{P}_y^0 \triangleq \frac{1}{|\{(\boldsymbol{x}_i, y_i) \in \mathcal{D}^T : y_i = y\}|} \sum_{(\boldsymbol{x}_i, y_i) \in \mathcal{D}^T, y_i = y} f(\boldsymbol{x}_i). \tag{3}$$

GPN repeatedly applies the following propagation procedure to update the prototypes in $\boldsymbol{P}_{\mathcal{Y}^T}$ for each class $y \in \mathcal{Y}^T$. At step-$t$, for each class $y \in \mathcal{Y}^T$, we firstly compute the aggregated messages from its neighbors $\mathcal{N}_y$ by a dot-product attention module $a(p, q)$, i.e.,

$$\boldsymbol{P}_{\mathcal{N}_y \to y}^{t+1} \triangleq \sum_{z \in \mathcal{N}_y} a(\boldsymbol{P}_y^t, \boldsymbol{P}_z^t) \times \boldsymbol{P}_z^t, \quad a(p, q) = \frac{\langle h_1(p), h_2(q) \rangle}{\|h_1(p)\| \times \|h_2(q)\|}. \tag{4}$$

where $h_1(\cdot)$ and $h_2(\cdot)$ are learnable transformations and their parameters $\Theta^{prop}$ are parts of the meta-learner parameters $\Theta$. To avoid the propagation to generate identical prototypes, we allow each class

$y$ to send its own last-step prototype $\boldsymbol{P}_y^t$ to itself, i.e., $\boldsymbol{P}_{y\to y}^{t+1} \triangleq \boldsymbol{P}_y^t$. Then we apply a gate $g$ making decisions of whether accepting messages $\boldsymbol{P}_{\mathcal{N}_y\to y}^{t+1}$ from its neighbors or message $\boldsymbol{P}_{y\to y}^{t+1}$ from itself, i.e.

$$\boldsymbol{P}_y^{t+1} \triangleq g\boldsymbol{P}_{y\to y}^{t+1} + (1-g)\boldsymbol{P}_{\mathcal{N}_y\to y}^{t+1}, \;\; g = \frac{\exp[\gamma\cos(\boldsymbol{P}_y^0, \boldsymbol{P}_{y\to y}^{t+1})]}{\exp[\gamma\cos(\boldsymbol{P}_y^0, \boldsymbol{P}_{y\to y}^{t+1})] + \exp[\gamma\cos(\boldsymbol{P}_y^0, \boldsymbol{P}_{\mathcal{N}_y\to y}^{t+1})]}, \;\; (5)$$

where $\cos(p,q)$ denotes the cosine similarity between two vectors $p$ and $q$, and $\gamma$ is a temperature hyper-parameter that controls the smoothness of the softmax function. To capture different types of relation and use them jointly for propagation, we apply $k$ modules of the above attentive and gated propagation (Eq. (4)-Eq. (5)) with untied parameters for $h_1(\cdot)$ and $h_2(\cdot)$ (as the multi-head attention in [25]) and average the outputs of the $k$ "heads", i.e.,

$$\boldsymbol{P}_y^{t+1} = \frac{1}{k}\sum_{i=1}^k \boldsymbol{P}_y^{t+1}[i], \tag{6}$$

where $\boldsymbol{P}_y^{t+1}[i]$ is the output of the $i$-th head and computed in the same way as $\boldsymbol{P}_y^{t+1}$ in Eq. (5). In GPN, we apply the above procedure to all $y \in \mathcal{Y}^T$ for $\mathcal{T}$ steps and the final prototype of class $y$ is given by

$$\boldsymbol{P}_y \triangleq \lambda \times \boldsymbol{P}_y^0 + (1-\lambda) \times \boldsymbol{P}_y^{\mathcal{T}}. \tag{7}$$

GPN can be trained in a life-long learning manner that relates tasks learned at different time steps by maintaining a memory of prototypes for all the classes on the graph that have been included in any previous task(s). This is especially helpful to learning the above propagation mechanism, because in practice it is common that many classes $y \in \mathcal{Y}^T$ do not have any neighbor in $\mathcal{Y}^T$, i.e., $\mathcal{N}_y \cap \mathcal{Y}^T = \emptyset$, so Eq. (4) cannot be applied and the propagation mechanism cannot be effectively trained. However, by initializing the prototypes of these classes to be the ones stored in memory, GPN is capable to apply propagation over all classes in $\mathcal{N}_y \cup \mathcal{Y}^T$ and thus relate any two classes on the graph, if there exists a path between them and all the classes on the path have prototypes stored in the memory.

## 3.3 Training Strategies

**Generating training tasks by subgraph sampling:** In meta-learning setting, we train GPN as a meta-learner on a set of training tasks. We can generate each task by sampling targeted classes $\mathcal{Y}^T$ using two possible methods: random sampling and snowball sampling [8]. The former randomly samples $N$ classes $\mathcal{Y}^T$ without using the graph, and they tend to be weakly related if the graph is sparse (which is often true). The latter selects classes sequentially: in each step, it randomly sample classes from the hop-$k_n$ neighbors of the previously selected classes, where $k_n$ is a hyper-parameter controlling how relative the selected classes are. In practice, we use a hybrid of them to cover more diverse tasks. Note $k_n$ also results in a trade-off: the classes selected into $\mathcal{Y}^T$ are close to each other when $k_n$ is small and they can provide strong training signals to learn the message passing; on the other hand, the tasks become hard because similar classes are easier to be confused.

**Building propagation pathways by maximum spanning tree:** During training, given a task $T$ defined on classes $\mathcal{Y}^T$, we need to decide the subgraph we apply the propagation procedure to, which can cover the classes $z \notin \mathcal{Y}^T$ but connected to some class $y \in \mathcal{Y}^T$ via some paths. Given that we apply $\mathcal{T}$ steps of propagation, it makes sense to add all the hop-1 to hop-$\mathcal{T}$ neighbors of every $y \in \mathcal{Y}^T$ to

---

**Algorithm 1** GPN Training

**Input:** $\mathcal{G} = (\mathcal{Y}, E)$, memory update interval $m$, propagation steps $\mathcal{T}$, total episodes $\tau_{total}$;
1: **Initialization:** $\Theta^{cnn}, \Theta^{prop}, \Theta^{fc}, \tau \leftarrow 0$;
2: **for** $\tau \in \{1, \cdots, \tau_{total}\}$ **do**
3:  **if** $\tau \mod m = 0$ **then**
4:   Update prototypes in memory by Eq. (3);
5:  **end if**
6:  Draw $\alpha \sim \text{Unif}[0, 1]$;
7:  **if** $\alpha < 0.9^{20\tau/\tau_t}$ **then**
8:   Train a classifier to update $\Theta^{cnn}$ with loss $\sum_{(\boldsymbol{x},y)\sim\mathcal{D}} -\log\Pr(y|\boldsymbol{x}; \Theta^{cnn}, \Theta^{fc})$;
9:  **else**
10:   Sample a few-shot task $T$ as in Sec. 3.3;
11:   Construct MST $\mathcal{Y}_{MST}^T$ as in Sec. 3.3;
12:   For $y \in \mathcal{Y}_{MST}^T$, compute $\boldsymbol{P}_y^0$ by Eq. (3) if $y \in T$, otherwise fetch $\boldsymbol{P}_y^0$ from memory;
13:   **for** $t \in \{1, \cdots, \mathcal{T}\}$ **do**
14:    For all $y \in \mathcal{Y}_{MST}^T$, concurrently update their prototypes $\boldsymbol{P}_y^t$ by Eq. (4)-(6);
15:   **end for**
16:   Compute $\boldsymbol{P}_y$ for $y \in \mathcal{Y}_{MST}^T$ by Eq.(7);
17:   Compute $\log\Pr(y|\boldsymbol{x}; \Theta^{cnn}, \Theta^{prop})$ by Eq. (2) for all samples $(\boldsymbol{x}, y)$ in task $T$;
18:   Update $\Theta^{cnn}$ and $\Theta^{prop}$ by minimizing $\sum_{(x,y)\sim\mathcal{D}^T} -\log\Pr(y|x; \Theta^{cnn}, \Theta^{prop})$;
19:  **end if**
20: **end for**

---

the subgraph. However, this might result in a large subgraph requiring costly propagation computa-

Table 2: Statistics of *tiered*ImageNet-Close and *tiered*ImageNet-Far for graph meta-learning, where #cls and #img denote the number of classes and images respectively.

| *tiered*ImageNet-Close | | | | | *tiered*ImageNet-Far | | | | |
|---|---|---|---|---|---|---|---|---|---|
| training | | test | | #img | training | | test | | #img |
| #cls | #img | #cls | #img | | #cls | #img | #cls | #img | |
| 773 | 100,320 | 315 | 45,640 | 145,960 | 773 | 100,320 | 26 | 12,700 | 113,020 |

tion. Hence, we further build a maximum spanning tree (MST) [13] (with edge weight defined by cosine similarity between prototypes from memory) $\mathcal{Y}^T_{MST}$ for the hop-$\mathcal{T}$ subgraph of $\mathcal{Y}^T$ as our "propagation pathways", and we only deply the propagation procedure on the MST $\mathcal{Y}^T_{MST}$. MST preserves the strongest relations to train the propagation and but significantly saves computations.

**Curriculum learning:** It is easier to train a classifier given sufficient training data than few-shot training data since the former is exposed to more supervised information. Inspired by auxiliary task in co-training [19], during early episodes of training[1], with high probability we learn from a traditional supervised learning task by training a linear classifier $\Theta^{fc}$ with input $f(\cdot)$ and update both the classifier and the representation model $f(\cdot)$. We gradually reduce the probability later on by using an annealed probability $0.9^{20\tau/\tau_t}$ so more training will target on few-shot tasks. Another curriculum we find helpful is to gradually reduce $\lambda$ in Eq. (7), since $\boldsymbol{P}^0_y$ often works better than $\boldsymbol{P}^{\mathcal{T}}_y$ in earlier episodes but with more training $\boldsymbol{P}^{\mathcal{T}}_y$ becomes more powerful. In particular, we set $\lambda = 1 - \tau/\tau_t$.

The complete training algorithm for GPN is given in Alg. 1. On image classification, we usually use CNNs for $f(\cdot)$. In GPN, the output of the meta-learner $\mathcal{M}(T; \Theta) = \{\boldsymbol{P}^y\}_{y \in \mathcal{Y}^T}$, i.e., the prototypes of class $y$ achieved in Eq. (7), and the meta-learner parameter $\Theta = \{\Theta^{cnn}, \Theta^{prop}\}$.

### 3.4 Applying a Pre-trained GPN to New Tasks

The outcomes of GPN training are the parameters $\{\Theta^{cnn}, \Theta^{prop}\}$ defining the GPN model and the prototypes of all the training classes stored in the memory. Given a new task $T$ with target classes $\mathcal{Y}^T$, we apply the procedure in lines 11-17 of Alg.1 to obtain the prototypes for all the classes in $\mathcal{Y}^T$ and the prediction probability of any possible test samples for the new task. Note that $\mathcal{Y}^T_{MST}$ can include training classes, so the test task can benefit from the prototypes of training classes in memory. However, this can directly work only when the graph already contains both the training classes and test classes in $T$. When test classes $\mathcal{Y}^T$ are not included in the graph, we apply an extra step at the beginning in order to connect test classes in $\mathcal{Y}^T$ to classes in the graph: we search for each test class's $k_c$ nearest neighbors among all the training prototypes in the space of $\boldsymbol{P}^0_y$, and add arcs from the test class to its nearest classes on the graph.

## 4 Experiments

In experiments, we conduct a thorough empirical study of GPN and compare it with several most recent methods for few-shot learning in 8 different settings of graph meta-learning on two datasets we extracted from ImageNet and specifically designed for graph meta-learning. We will briefly introduce the 8 settings below. First, the similarity between test tasks and training tasks may influence the performance of a graph meta-learning method. We can measure the distance/dissimilarity of a test class to a training class by the length (i.e., the number of edges) of the shortest path between them. Intuitively, propagation brings more improvement when the distance is smaller. For example, when test class "laptop" has nearest neighbor "electronic devices" in training classes, the prototype of electronic devices can provide more related information during propagation when generating the prototype for laptop and thus improve the performance. In contrast, if the nearest neighbor is "device", then the help by doing prototype propagation might be very limited. Hence, we extract two datasets from ImageNet with different distance between test classes and training classes. Second, as we mentioned in Sec. 3.4, in real-world problems, it is possible that test classes are not included in the graph during training. Hence, we also test GPN in the two scenarios (denoted by GPN+ and GPN) when the test classes have been included in the graph or not. At last, we also evaluate GPN with two different sampling methods as discussed in Sec. 3.3. The combination of the above three options finally results in 8 different settings under which we test GPN and/or other baselines.

### 4.1 Datasets

**Importance.** We built two datasets with different distance/dissimilarity between test classes and training classes, i.e., *tiered*ImageNet-Close and *tiered*ImageNet-Far. To the best of our knowledge, they are the first two benchmark datasets that can be used to evaluate graph meta-learning methods for few-shot learning. Their importance are: 1) The proposed datasets (and the method to generate datasets) provide benchmarks for the novel graph meta-learning problem, which is practically important since it uses the normally available graph information to improve the few-shot learning performance, and is a more general challenge since it covers classification tasks of any classes from the graph rather than only the finest ones. 2) On these datasets, we empirically justified that the relation among tasks (reflected by class connections on a graph) is an important and easy-to-reach source of meta-knowledge which can improve meta-learning performance but has not been studied by previous works. 3) The proposed datasets also provide different graph morphology to evaluate the meta knowledge transfer through classes in different methods: Every graph has 13 levels and covers $\sim$ 800 classes/nodes and it is flexible to sample a subgraph or extend to a larger graph using our released code. So we can design more and diverse learning tasks for evaluating meta-learning algorithms.

**Details.** The steps for the datasets generation procudure are as follows: 1) Build directed acyclic graph (DAG) from the root node to leaf nodes (a subset of ImageNet classes [22]) according to WordNet. 2) Randomly sample training and test classes on the DAG that satisfy the pre-defined minimum distance conditions between the training classes and test classes. 3) Randomly sample images for every selected class, where the images of a non-leaf class are sampled from their descendant leaf classes, e.g. the animal class has images sampled from dogs, birds, etc., all with only a coarse label "animal". The two datasets share the same training tasks and we make sure that there is no overlap between training and test classes. Their difference is at the test classes. In *tiered*ImageNet-Close, the minimal distance between each test class to a training class is 1$\sim$4, while the minimal distance goes up to 5$\sim$10 in *tiered*ImageNet-Far. The statistics for *tiered*ImageNet-Close and *tiered*ImageNet-Far are reported in Table 2.

### 4.2 Experiment Setup

We used $k_n = 5$ for snowball sampling in Sec. 3.3. The training took $\tau_{total} =350k$ episodes using Adam [12] with an initial learning rate of $10^{-3}$ and weight decay $10^{-5}$. We reduced the learning rate by a factor of $0.9\times$ every $10k$ episodes starting from the $20k$-th episode. The batch size for the auxiliary task was $128$. For simplicity, the propagation steps $\mathcal{T} = 2$. More steps may result in higher performance with the price of more computations. The interval for memory update is $m = 3$ and the the number of heads is $5$ in GPN. For the setting that test class is not included in the original graph, we connect it to the $k_c = 2$ nearest training classes. We use linear transformation for $g(\cdot)$ and $h(\cdot)$. For fair comparison, we used the same backbone ResNet-08 [9] and the same setup of the training tasks, i.e., $N$-way-$K$-shot, for all methods in our experiments. Our model took approximately 27 hours on one TITAN XP for the 5-way-1-shot learning. The computational cost can be reduced by updating the memory less often and applying fewer steps of propagation.

### 4.3 Results

**Selection of baselines.** We chose meta-learning baselines that are mostly related to the idea of metric/prototype learning (Prototypical Net [24], PPN [15] and Closer Look [3]) and prototype propagation/message passing (PPN [15]). We also tried to include the most recent meta-learning methods published in 2019, e.g., Closer Look [3] and PPN [15].

The results for all the methods on *tiered*ImageNet-Close are shown in Table 3 for tasks generated by random sampling, and Table 4 for tasks generated by snowball sampling. The results on *tiered*ImageNet-Far is shown in Table 5 and Table 6 with the same format. GPN has compelling generalization to new tasks and shows improvements on various datasets with different kinds of tasks. GPN performs better with smaller distance between the training and test classes, and achieves up to $\sim$8% improvement with random sampling and $\sim$6% improvement with snowball sampling compared to baselines. Knowing the connections of test classes to training classes in the graph (GPN+) is more helpful on *tiered*ImageNet-Close, which brings 1$\sim$2% improvement on average compared to the situation without hierarchy information (GPN). The reason is that *tiered*ImageNet-Close contains more important information about class relations that can be captured by GPN+. In contrast, on

*tiered*ImageNet-Far, the graph only provides weak/far relationship information, thus GPN+ is not as helpful as it shows on *tiered*ImageNet-Close.

Table 3: Validation accuracy (mean±CI%95) on 600 test tasks achieved by GPN and baselines on *tiered*ImageNet-**Close** with few-shot tasks generated by **random sampling**.

| Model | 5way1shot | 5way5shot | 10way1shot | 10way5shot |
|---|---|---|---|---|
| Prototypical Net [24] | 42.87±1.67% | 62.68±0.99% | 30.65±1.15% | 48.64±0.70% |
| GNN [6] | 42.33±0.80% | 59.17±0.69% | 30.50±0.57% | 44.33±0.72% |
| Closer Look [3] | 35.07±1.53% | 47.48±0.87% | 21.58±0.96% | 28.01±0.40% |
| PPN [15] | 41.60±1.59% | 63.04±0.97% | 28.48±1.09% | 48.66±0.70% |
| GPN | 48.37±1.80% | 64.14±1.00% | 33.23±1.05% | 50.50±0.70% |
| GPN+ | **50.54±1.67%** | **65.74±0.98%** | **34.74±1.05%** | **51.50±0.70%** |

Table 4: Validation accuracy (mean±CI%95) on 600 test tasks achieved by GPN and baselines on *tiered*ImageNet-**Close** with few-shot tasks generated by **snowball sampling**.

| Model | 5way1shot | 5way5shot | 10way1shot | 10way5shot |
|---|---|---|---|---|
| Prototypical Net [24] | 35.27±1.63% | 52.60±1.17% | 26.08±1.04% | 41.48±0.76% |
| GNN [6] | 36.50±1.03% | 52.33±0.96% | 27.67±1.01% | 40.67±0.90% |
| Closer Look [3] | 34.07±1.63% | 47.48±0.87% | 21.02±0.99% | 33.70±0.44% |
| PPN [15] | 36.50±1.62% | 52.50±1.12% | 27.18±1.08% | 40.97±0.77% |
| GPN | 39.56±1.70% | 54.35±1.11% | 27.99±1.09% | 42.50±0.76% |
| GPN+ | **40.78±1.76%** | **55.47±1.41%** | **29.46±1.10%** | **43.76±0.74%** |

Table 5: Validation accuracy (mean±CI%95) on 600 test tasks achieved by GPN and baselines on *tiered*ImageNet-**Far** with few-shot tasks generated by **random sampling**.

| Model | 5way1shot | 5way5shot | 10way1shot | 10way5shot |
|---|---|---|---|---|
| Prototypical Net [24] | 44.30±1.63% | 61.01±1.03% | 30.63±1.07% | 47.19±0.68% |
| GNN [6] | 43.67±0.69% | 59.33±1.04% | 30.17±0.47% | 43.00±0.66% |
| Closer Look [3] | 42.27±1.70% | 58.78±0.94% | 22.00±0.99% | 32.73±0.41% |
| PPN [15] | 43.63±1.59% | 60.20±1.02% | 29.55±1.09% | 46.72±0.66% |
| GPN | **47.54±1.68%** | **64.20±1.01%** | **31.84±1.10%** | 48.20±0.69% |
| GPN+ | **47.49±1.67%** | **64.14±1.02%** | **31.95±1.15%** | **48.65±0.66%** |

Table 6: Validation accuracy (mean±CI%95) on 600 test tasks achieved by GPN and baselines on *tiered*ImageNet-**Far** with few-shot tasks generated by **snowball sampling**.

| Model | 5way1shot | 5way5shot | 10way1shot | 10way5shot |
|---|---|---|---|---|
| Prototypical Net [24] | 43.57±1.67% | 62.35±1.06% | 29.88±1.11% | 46.48±0.70% |
| GNN [6] | 44.00±1.36% | 62.00±0.66% | 28.50±0.60% | 46.17±0.74% |
| Closer Look [3] | 38.37±1.57% | 54.64±0.85% | 30.40±1.09% | 33.72±0.43% |
| PPN [15] | 42.40±1.63% | 61.37±1.05% | 28.67±1.01% | 46.02±0.61% |
| GPN | **47.74±1.76%** | **63.53±1.03%** | **32.94±1.16%** | **47.43±0.67%** |
| GPN+ | **47.58±1.70%** | **63.74±0.95%** | **32.68±1.17%** | **47.44±0.71%** |

## 4.4 Visualization of Prototypes Achieved by Propagation

We visualize the prototypes before (i.e., the ones achieved by Prototypical Networks) and after (GPN) propagation in Figure. 3. Propagation tends to reduce the intra-class variance by producing similar prototypes for the same class in different tasks. The importance of reducing intra-class variance in few-shot learning has also been mentioned in [3, 7]. This result indicates that GPN is more powerful to find the relations between different tasks, which is essential for meta-learning.

## 4.5 Ablation Study

In Table 7, we report the performance of many possible variants of GPN. In particular, we change the task generation methods, propagation orders on the graph, training strategies, and attention

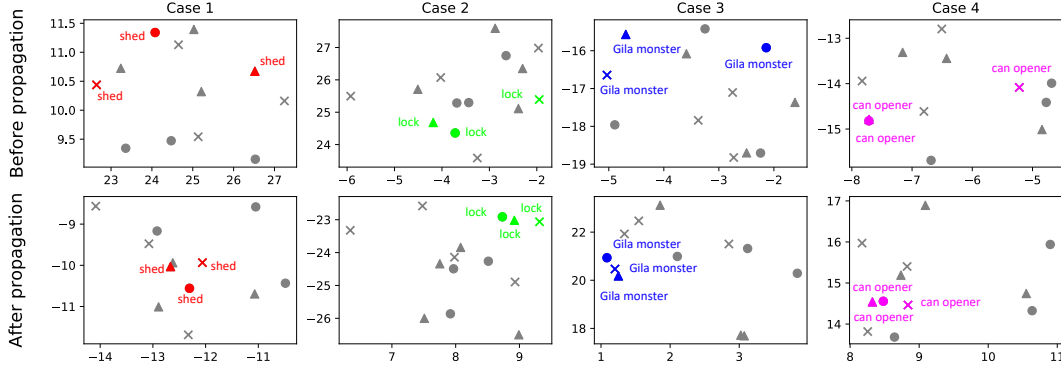

Figure 3: Prototypes before (top row) and after GPN propagation (bottom row) on *tiered*ImageNet-Close by random sampling for 5-way-1-shot few-shot learning. The prototypes in top row equal to the ones achieved by prototypical network. Different tasks are marked by a different shape (∘/×/△), and classes shared by different tasks are highlighted by non-grey colors. It shows that GPN is capable to map the prototypes of the same class in different tasks to the same region. Comparing to the result of prototypical network, GPN is more powerful in relating different tasks.

Table 7: Validation accuracy (mean±CI%95) of GPN variants on *tiered*ImageNet-Close for 5-way-1-shot tasks. Original GPN's choices are in **bold** fonts. Details of the variants are given in Sec. 4.5.

| Task Generation | | | Propagation Mechanism | | | | | Training | | Model | | | ACCURACY |
|---|---|---|---|---|---|---|---|---|---|---|---|---|---|
| **SR-S** | S-S | R-S | **N→C** | F→C | C→C | B→P | M→P | **AUX** | **MST** | **M-H** | M-A | A-A | |
| | ✓ | | ✓ | | | | | ✓ | ✓ | ✓ | ✓ | | 46.20±1.70% |
| | | ✓ | ✓ | | | | | ✓ | ✓ | ✓ | ✓ | | 49.33±1.68% |
| ✓ | | | | ✓ | | | | ✓ | ✓ | ✓ | ✓ | | 42.60±1.61% |
| ✓ | | | | | ✓ | | | ✓ | ✓ | ✓ | ✓ | | 37.90±1.50% |
| ✓ | | | | | | ✓ | | ✓ | ✓ | ✓ | ✓ | | 47.90±1.72% |
| ✓ | | | | | | | ✓ | ✓ | ✓ | ✓ | ✓ | | 46.90±1.78% |
| ✓ | | | ✓ | | | | | | ✓ | ✓ | ✓ | | 41.87±1.72% |
| ✓ | | | ✓ | | | | | ✓ | | ✓ | ✓ | | 45.83±1.64% |
| ✓ | | | ✓ | | | | | ✓ | ✓ | | ✓ | | 49.40±1.69% |
| ✓ | | | ✓ | | | | | ✓ | ✓ | ✓ | | ✓ | 46.74±1.71% |
| ✓ | | | ✓ | | | | | ✓ | ✓ | ✓ | ✓ | | **50.54±1.67%** |

modules, in order to make sure that the choices we made in the paper are the best for GPN. For task generation, GPN adopts both random and snowball sampling (**SR-S**), which performs better than snowball sampling only (S-S) or random sampling only (R-S). We also compare different choices of propagation directions, i.e., **N→C** (messages from neighbors, used in the paper), F→C (messages from parents) and C→C (messages from children). B→P follows the ideas of belief propagation [21] and applies forward propagation for $\mathcal{T}$ steps along the hierarchy and then applies backward propagation for $\mathcal{T}$ steps. M→P applies one step of forward propagation followed by a backward propagation step and repeat this process for $\mathcal{T}$ steps. The propagation order introduced in the paper, i.e., N→C, shows the best performance. It shows that the auxiliary tasks (**AUX**), maximum spanning tree (**MST**) and multi-head (**M-H**) are important reasons for better performance. We compare the multi-head attention (**M-H**) using multiplicative attention (**M-A**) and using additive attention (A-A), and the former has better performance.

## Acknowledgements

This research was funded by the Australian Government through the Australian Research Council (ARC) under grants 1) LP160100630 partnership with Australia Government Department of Health and 2) LP150100671 partnership with Australia Research Alliance for Children and Youth (ARACY) and Global Business College Australia (GBCA). We also acknowledge the support of NVIDIA Corporation and Google Cloud with the donation of GPUs and computation credits.

## Footnotes

[1]We update GPN in each episode $\tau$ on a training task $T$, and train GPN for $\tau_{total}$ episodes.

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

# A    Visualization Results

## A.1    Prototype Hierarchy

We show more visualizations for the hierarchy structure of the training prototypes in Figure. 4.

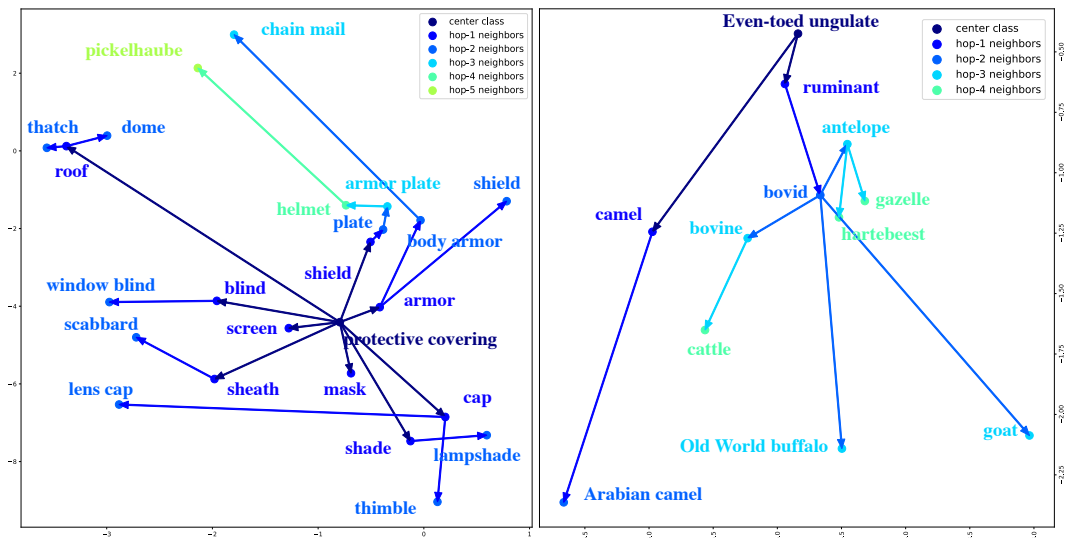

Figure 4: Visualization of the hierarchy structure of subgraphs from the training class prototypes transformed by t-SNE.

## A.2    Prototypes Before and After Propagation

We show more visualization examples for the comparison of the prototypes learned before (Prototypical Networks) and after propagation (GPN) in Figure. 5.

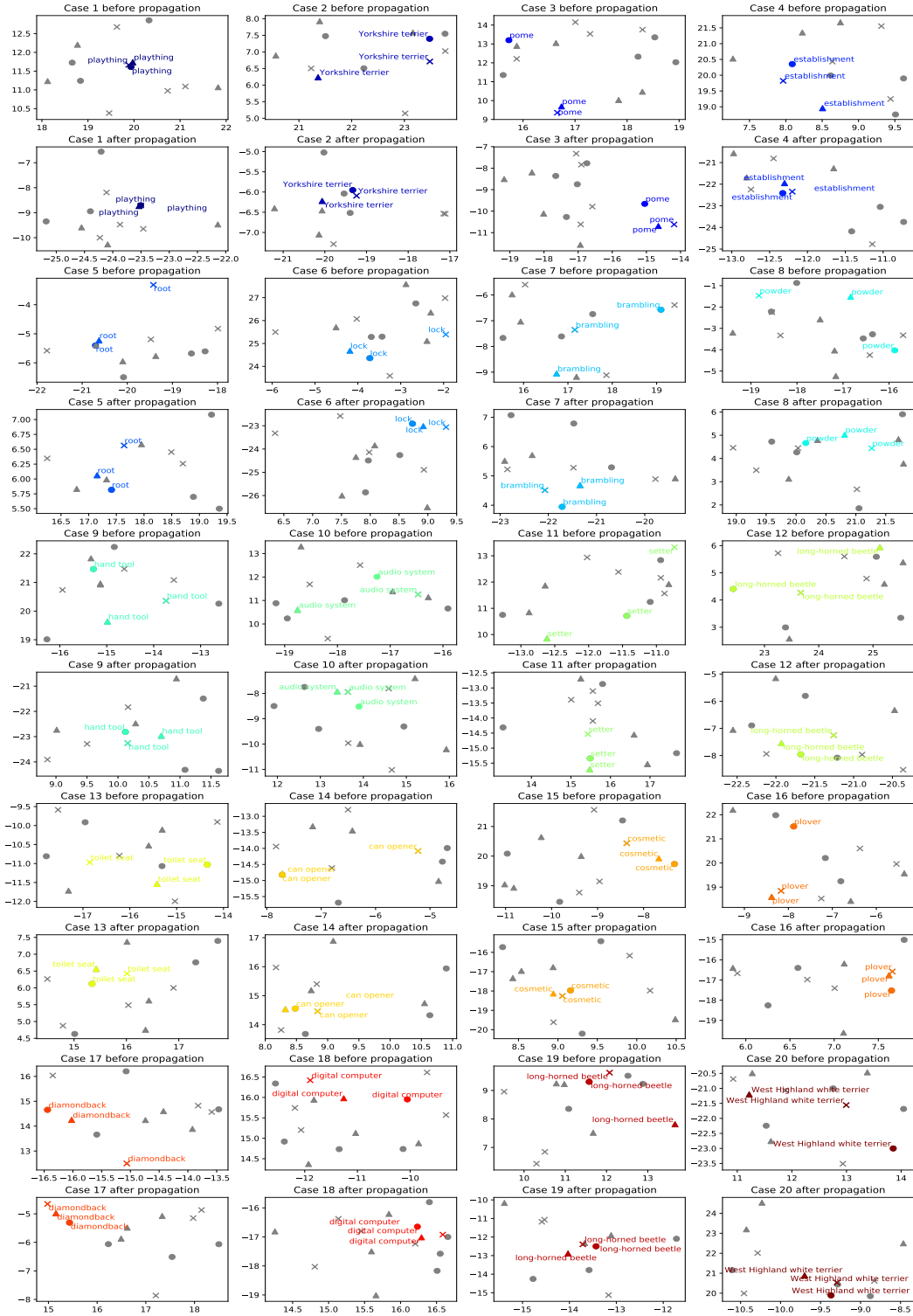

Figure 5: Prototypes before and after GPN propagation on *tiered*ImageNet-Close by random sampling for 5-way-1-shot few-shot learning. The prototypes in top row equal to the ones achieved by prototypical network. Different tasks are marked by a different shape (○/×/△), and classes shared by different tasks are highlighted by non-grey colors. It shows that GPN is capable to map the prototypes of the same class in different tasks to the same region. Comparing to the result of prototypical network, GPN is more powerful in relating different tasks.

