[Reviews · NeurIPS 2019]

Reviewer 1



Originality: While the components of the algorithm (i.e., message passing, attention, graph-embeddings etc.) are not new, this particular combination is novel, and required a nontrivial effort deploy effectively. Quality: The work is technically sound, and the evaluation criteria are robust and extensive (though I am curious how this compares with MAML style metalearning techniques). Clarity: The paper is fairly clear, but could be improved considerably with code for implementation. (There are also a handful of minor spelling errors throughout). Graph-based methods and message-passing (in particular) are relatively uncommon, thus providing more detail could only help the traction of the work. Significance: The work seems significant (as evidenced by the clear advantage on the provided baselines), but I will admit to a small amount of bewilderment at the sheer number of separate ways in which people have chosen to test few-shot metalearning systems. This is partially the community's fault (for failure to align on a clear baseline), but it would help it the authors spent a bit more time contextualizing their choice of baseline.

Reviewer 2



It's a novel combination of well-known methods. The submission is technically sound. However, it is not very well written with a clustering of notations and difficult to follow all the equations and algorithms. The paper is not very well organized without conclusion. The proposed method achieves state-of-the-art accuracy on the newly introduced datasets. Not sure it is a fair comparison with the baseline methods as other published methods are not specifically designed for these tasks as well.

Reviewer 3



1. Originality: The framework proposed by the paper is a novel model (GPN model) based on the prototypical network. The prototype embeddings are refined iteratively through the current presentation and neighboring prototypes from similar tasks using a gating mechanism. The refinement process is similar to the multi-head mechanism but is new and unique. 2. Quality & Clarity: The paper overall has a clear and concise description of the methodology, which is empirically supported by the experimental results. 3. Significance: The paper has compared the proposed GPN model with several state-of-the-art few-shot learning methods including the prototypical net baseline. The evaluation process on ImageNet is reasonable which demonstrates the effectiveness of the method on both closely-related and distant tasks. However, as the data are created by the authors and not released, it may be hard to reproduce the results based on the current information in the submission. Minors: 1. In Line 173, the authors mention it is possible to use the history prototypes for better training performance for GPN model. I am wondering how this general technique can be applied to and influence other baselines?

[Author Response · NeurIPS 2019]

We appreciate the reviewers' time, efforts, and valuable suggestions! We will address the suggestions in the next version.
We open-source our implementation and the code to generate the proposed datasets via an anonymous link given below.
We will do our best to improve the clarity and organization, and add explanation of background and dataset generation.

————————————————————————**To Reviewer 1**————————————————————————-

**Improvements: Code for implementation.** Our code for GPN and dataset generation are now released at
https://github.com/Anonymous-Code-CS/GPN.
**I am curious how this compares with MAML style meta-learning techniques.** MAML aims to learn an initializa-
tion point instead of a similarity metric and thus is not a natural choice for graph meta-learning. Designing a MAML
style method for graph meta-learning would be an interesting topic that we will further study in future work.
**It would help if the authors spent a bit more time contextualizing the choice of baseline.** We chose meta-learning
baselines that are mostly related to the idea of metric/prototype learning (Prototypical Net [24], PPN [15] and Closer
Look [3]) and prototype propagation/message passing (PPN [15]). We also tried to include the most recent meta-learning
methods published in 2019, e.g., Closer Look [3] and PPN [15].

————————————————————————**To Reviewer 2**————————————————————————-

**Improvements: Clarity and organization.** We will add a notation table, improve clarity, and add a conclusion.
**Improvements: More explanation on the importance of the newly introduced datasets.** 1) The proposed datasets
(and the method to generate datasets) provide benchmarks for the novel graph meta-learning problem, which is practically
important since it uses the normally available graph information to improve the few-shot learning performance, and is a
more general challenge since it covers classification tasks of any classes from the graph rather than only the finest ones.
2) On these datasets, we empirically justified that the relation among tasks (reflected by class connections on a graph)
is an important and easy-to-reach source of meta-knowledge which can improve meta-learning performance but has
not been studied by previous works. 3) The proposed datasets also provide different graph morphology to evaluate the
meta knowledge transfer through classes in different methods: Every graph has 13 levels and covers $\sim$ 1k classes/nodes
and it is flexible to sample a subgraph or extend to a larger graph using our released code. So we can design more and
diverse learning tasks for evaluating meta-learning algorithms. We will add more discussions to the paper.
**Not sure it is a fair comparison with the baseline methods as other published methods are not specifically**
**designed for these tasks as well.** 1) Although the graph meta-learning task is novel, we tried to select baselines that
have the similar capabilities as our method, e.g., learning prototypes, similarity metrics and/or propagation on graph. 2)
The datasets are not specifically designed for our method: they are generated by random sampling, and we reported the
comparison results with four different ways of generating datasets in Table 2-5. 3) The learning tasks in our evaluation
are more general since they can be classifications over any classes on the graph, while the other baselines are designed
for a special case that all the classes only come from the leaf nodes (i.e., the finest classes) on the graph.

————————————————————————**To Reviewer 3**————————————————————————-

**Improvements: It would be helpful to specify the details for the dataset generation procedure for reproduction.**
The code for GPN and dataset generation are now released at https://github.com/Anonymous-Code-CS/GPN for
reproducing purpose. In the paper, we will also add detailed steps of dataset extraction from ImageNet and WordNet:
1) Build directed acyclic graph (DAG) from the root node to leaf nodes (a subset of ImageNet classes) according to
WordNet. 2) Randomly sample training and test classes on the DAG that satisfy the pre-defined minimum distance
conditions between the training classes and test classes. 3) Randomly sample images for every selected class, where the
images of a non-leaf class are sampled from their descendant leaf classes, e.g. the animal class has images sampled
from dogs, birds, etc., all with only a coarse label "animal".
**Improvements: Discuss/compare the general technique on other baseline models. In Line 173, the authors**
**mention it is possible to use the history prototypes for better training performance for GPN model. I am**
**wondering how this general technique can be applied to other baselines?** 1) The main purpose of using history
prototypes is to save the computation of calculating all the prototypes in Eq.3 for each episode. As explained above Line
173, we need to relate two distant classes in a few-shot task by propagation from one class's prototype to the other via
their neighboring classes, which requires to know their neighbor classes' prototypes. If some neighbor classes are not in
the classes of the few-shot task, we use their pre-computed history prototypes instead of the ones computed by Eq.3 for
better efficiency. For better trade-off between efficiency and prototype freshness, as in Line 3-5 of Algorithm 1, we
apply Eq.3 to update all prototypes every $m$ episodes. 2) Because of above, only the class-prototype propagation-based
baselines have the similar efficiency-freshness trade-off problem, i.e., only PPN in our paper (we have applied it). For
baselines without propagation, we can always compute the prototype per class in a few-shot task by averaging over the
features of samples belonging to that class, so it is not necessary (and might be worse) to use history prototypes in this
case. 3) It might be helpful to developing prototype propagation methods for life-long learning task.

[Meta-Review · NeurIPS 2019]

The reviewers agree that the proposed GPN is a novel combination of several components from the literature and represents a good contribution to the meta learning community. Please be sure to include a notation table as requested by one reviewer, along with the additional explanations/clarifications provided in the rebuttal.